# Genetic Alterations in Mitochondrial DNA Are Complementary to Nuclear DNA Mutations in Pheochromocytomas

**DOI:** 10.3390/cancers14020269

**Published:** 2022-01-06

**Authors:** Mouna Tabebi, Małgorzata Łysiak, Ravi Kumar Dutta, Sandra Lomazzi, Maria V. Turkina, Laurent Brunaud, Oliver Gimm, Peter Söderkvist

**Affiliations:** 1Department of Biomedical and Clinical Sciences (BKV), Linköping University, 581 83 Linköping, Sweden; malgorzata.lysiak@liu.se (M.Ł.); rdutta2@bwh.harvard.edu (R.K.D.); maria.turkina@liu.se (M.V.T.); oliver.gimm@liu.se (O.G.); peter.soderkvist@liu.se (P.S.); 2Centre de Ressources Biologiques (CRB) Lorraine, CHRU de Nancy, 54511 Nancy, France; s.lomazzi@chu-nancy.fr; 3Department of Gastrointestinal, Metabolic, and Oncology Surgery (CVMC), Section of Metabolic, Endocrine, and Thyroid Surgery (UMET) at the CHRU Nancy, Hôpital de Brabois, Inserm U1256, Faculté de Médecine, Université de Lorraine, 54511 Vandoeuvre-les-Nancy, France; l.brunaud@chru-nancy.fr; 4Department of Surgery and Department of Biomedical and Clinical Sciences (BKV), Linköping University, 581 83 Linköping, Sweden; 5Clinical Genomics Linköping, Science for Life Laboratory, Linköping University, 581 83 Linköping, Sweden

**Keywords:** mitochondrial DNA, genetic alterations, pheochromocytomas and paragangliomas

## Abstract

**Simple Summary:**

Mitochondrial DNA (mtDNA) alterations have been reported to play important roles in cancer development and metastasis. However, there is scarce information about pheochromocytomas and paragangliomas (PCCs/PGLs) formation. To determine the potential roles of mtDNA alterations in PCCs/PGLs, we analyzed a panel of 26 nuclear susceptibility genes and the entire mtDNA sequence of 77 human tumors, using NGS. We also performed an analysis of copy-number alterations, large mtDNA deletion, and gene/protein expression. Our results revealed that 53.2% of the tumors harbor a mutation in the susceptibility genes and 16.9% harbor complementary mitochondrial mutations. Large deletions and depletion of mtDNA were found in 26% and 87% of tumors, respectively, accompanied by a reduced expression of the mitochondrial biogenesis markers (PCG1α, NRF1, and TFAM). Furthermore, P62 and LC3a gene expression suggested increased mitophagy, which is linked to mitochondrial dysfunction. These finding suggest a complementarity and a potential contributing role in PCCs/PGLs tumorigenesis.

**Abstract:**

Background: Somatic mutations, copy-number variations, and genome instability of mitochondrial DNA (mtDNA) have been reported in different types of cancers and are suggested to play important roles in cancer development and metastasis. However, there is scarce information about pheochromocytomas and paragangliomas (PCCs/PGLs) formation. Material: To determine the potential roles of mtDNA alterations in sporadic PCCs/PGLs, we analyzed a panel of 26 nuclear susceptibility genes and the entire mtDNA sequence of seventy-seven human tumors, using next-generation sequencing, and compared the results with normal adrenal medulla tissues. We also performed an analysis of copy-number alterations, large mtDNA deletion, and gene and protein expression. Results: Our results revealed that 53.2% of the tumors harbor a mutation in at least one of the targeted susceptibility genes, and 16.9% harbor complementary mitochondrial mutations. More than 50% of the mitochondrial mutations were novel and predicted pathogenic, affecting mitochondrial oxidative phosphorylation. Large deletions were found in 26% of tumors, and depletion of mtDNA occurred in more than 87% of PCCs/PGLs. The reduction of the mitochondrial number was accompanied by a reduced expression of the regulators that promote mitochondrial biogenesis (PCG1α, NRF1, and TFAM). Further, P62 and LC3a gene expression suggested increased mitophagy, which is linked to mitochondrial dysfunction. Conclusion: The pathogenic role of these finding remains to be shown, but we suggest a complementarity and a potential contributing role in PCCs/PGLs tumorigenesis.

## 1. Introduction

Pheochromocytomas (PCCs) and paragangliomas (PGLs) are rare endocrine tumors that arise from the adrenal medulla and in paraganglia of the autonomous nervous system, respectively. Up to 40% of pheochromocytomas are hereditary and caused by germline mutations in well-known cancer susceptibility genes (*SDHx*, *VHL*, *EGLN1/PHD2*, *EPAS1/HIF2A*, *KIF1Bβ*, *MAX*, *MEN1*, *NF1*, *RET*, *TMEM127*, *RAS*, *NF1*, *FGFR1*, *ATRX*, etc.) which are involved in interconnecting pathways [1,2]. Sporadic pheochromocytomas can have somatic mutations in one of these hereditary genes, indicating that they may be the main drivers. Together, germline and somatic mutations in susceptibility genes are found in approximately 60% of all PCCs/PGLs [3,4], but the cause of many sporadic PCCs/PGLs is still unknown [5]. With regard to gene-expression patterns, at least two main signatures are distinguished, clusters 1 and 2. Cluster 1 tumors (having mutations in, for example, *VHL* and *SDHA/B/C/D/AF2*) are characterized by increased expression of genes involved in (pseudo) hypoxia, mitochondrial electron transport chain, Krebs cycle, cell proliferation, and angiogenesis. Cluster 2 tumors (having mutations in, for example, *RET*, *NF1*, *MAX*, and *TMEM127*) are characterized by an increased expression of genes involved in cell proliferation, protein synthesis, and kinase signaling. Recently, a set of 46 genes was used by Flynn et al. (2016) to develop a Pheo-type gene-expression profiling, where the samples were separated into two main subtypes: pseudohypoxia and RTK/Ras driven tumors. Pseudohypoxia clustering is defined by overexpression of the transcription factor, *TRIB3*, and the cell adhesion gene, *DSP*, for SDHx tumors and overexpression of angiogenesis-associated genes (i.e., *ETS1*, *CD34*, and *AQP1*) for VHL tumors. RTK clustering is based on the overexpression of a chromaffin cell differentiation signature (i.e., *PNMT* and *RET*) within *NF1*, *RET*, *HRAS*, and *TMEM127* tumors and underexpression of genes associated with the MAP kinase pathway (i.e., *SPRY4* and *DUSP6*) and metabolism (i.e., *CSGALNACT1* and *PFKFB3*) for MAX-like tumors [6].

The nuclear genes representing cluster 1 have a main role in the pathogenesis of PCCs/PGLs and other types of cancer by affecting the cellular metabolism and mitochondrial function. However, cancer pathogenesis commonly involves the accumulation of several genetic alterations, not only in the nuclear genome but also in the mitochondrial genomes. In the past few years, several studies have reported a significant number of mtDNA mutations, including point mutations, deletions, and insertions, in a variety of human malignancies, including head and neck [7], lung [8], and breast [9]. It is believed that mtDNA mutations and copy-number alterations are associated with tumor malignancies, because of the high levels of reactive oxygen species (ROS) produced during oxidative phosphorylation (OXPHOS) and oxidative mtDNA damage, along with a less efficient mtDNA repair system [10,11]. The depletion of mtDNA has also been identified in various cancers [12] and has been suggested as a potential prognostic marker in cancers of the bladder, breast, kidney, head and neck, esophagus, and liver [13,14].

Recent publications have shown that catecholamine metabolism is fundamental to mtDNA integrity and mitochondrial function [15,16], but mitochondrial genome analysis has not been investigated in PCCs/PGLs yet.

In this study, we investigated mutational status/mutations in a panel of 26 susceptibility genes and the mitochondrial genome in PCCs/PGLs tumors by performing an extensive analysis of mutations, using next-generation sequencing, mitochondrial copy-number alterations, large deletions, and gene and protein expression.

## 2. Materials and Methods

### 2.1. Patients and Tumors

This study includes 74 pheochromocytomas and 3 paragangliomas from 77 patients having a sporadic disease, i.e., without family history and any syndromic features (see Appendix A). Fifty-four tumors were obtained from the Brabois Hospital, Nancy, France, and the remaining 23 tumors were obtained from Linkoping University Hospital, Sweden. All tumors were fresh-frozen. As a control, peripheral blood was obtained from 22 Swedish patients, and normal fresh-frozen tissues were obtained from 50 French patients. All normal tissues from the adrenal medulla were carefully dissected from the tumors during the surgery, at a distance normally larger than 1 mm, followed by tumor dissection and macroscopic aspect. All samples were histologically confirmed as PCCs/PGLs using World Health Organization (WHO) criteria and the distinction between malignant and benign PCCs/PGLs was evaluated according to the Endocrine Society Clinical Guidelines Subcommittee (CGS) criteria [17]. All tissue samples were handled following a Standard Operation Procedure. In short, resection specimens were stored in labeled cryovials and snap-frozen in liquid nitrogen. The time laps between sample resection and freezing were less than 15 min. All samples were collected and studied (Appendix B) with informed consent and approval from the local ethics committees (Dnr 2010/40-31, Dnr 2015/175-32 Linköping University, Sweden; DR-2016-346, CHRU de Nancy, France).

### 2.2. Next-Generation Sequencing

Probes for targeted sequencing with Twist custom target enrichment kit were designed to cover 26 susceptibility genes (SDHA, SDHB, SDHC, SDHD, SDHAF2, KIF1B, CSDE1, ARNT, EGLN1, FH, RET, VHL, HRAS, MAX, NF1, UBTF, TCF4, MYCN, EPAS1, TMEM127, BAP1, BRAF, FGFR1, SCAI, ATRX, and D2HGDH). The 26 genes are the most frequently mutated genes in both sporadic and hereditary PCCs/PGLs (see Appendix A [18,19,20,21,22,23,24,25,26,27,28,29,30]). The DNA library was prepared according to the manufacturer’s protocol, and all 77 samples included in one library pool were sequenced on a NextSeq500 sequencer (Illumina), using 2 × 75 bp paired-end reads. The alignment of the NGS data to the human reference genome (hg19) and the variant calling, including the annotation, were performed by using “LRM” (Local Run Manager V2) software provided by Illumina. The variant classification was based on VarSome (https://varsome.com/) (accessed on 24 September 2021) that uses the American College of Medical Genetics and Genomics guidelines [31,32].

### 2.3. Mitochondrial DNA Analysis

#### 2.3.1. Mitochondrial Genome Sequencing

DNA concentration was measured with Quantus™ Fluorometer (Sigma). After fragmentation with the Bioruptor Plus (Diagenode) (13 cycles 30″/30″), library preparation was performed for all 77 samples via Accel-NGS 2S Plus DNA Library kit (Swift Biosciences) according to the protocol. Hybridization capture was performed with myBaits Mito panels (Arbor Biosciences) with bait concentration appropriate for recovering full mtDNA sequences, following the manufacturer’s recommendation. Sequencing was performed on the Illumina MiSeq system (paired-end 2 × 150 bp). The data analysis was performed by using two software programs from Illumina: mtDNA Variant Processor V1.0.0 (processes FASTQ files directly from MiSeq instrument and generates VCF files) and mtDNA Variant Analyzer V1.0.0 (displays the VCF output for visualization and generate report in an .xls format). The mitochondrial haplogroup analysis was performed by using MitoTool database (http://www.mitotool.org/genome.html) (accessed on 23 September 2021) (Appendix C).

#### 2.3.2. Mitochondrial Copy-Number Variation and DNA Large Deletion

Mitochondrial DNA-copy number analysis was performed by digital droplet PCR (ddPCR), as described previously by Memon et al. [33], using probes targeting the mitochondrially encoded NADH dehydrogenase 1 (*MT-ND1*) (assay ID: dHsaCPE5029120, Bio-Rad) gene labeled with FAM fluorophore (mitochondrial DNA) and the eukaryotic translation initiation factor 2C, 1 (*EIF2C1*) (assay ID: dHsaCP1000002, Bio-Rad), and ribonuclease P/MRP 30 kDa subunit (*RPP30*) (assay ID: dHsaCP1000485, Bio-Rad) genes (nuclear DNA) labeled with HEX. Normal adrenal medulla tissues from the French cohort were used as a control group to estimate the mitochondrial copy-number variation in both tumor cohorts.

Long-range PCR was performed to detect mtDNA deletions, using Long PCR Enzyme Mix, according to manufacturer’s instructions (LA Taq DNA polymerase # RR002A, Takara). To amplify the major arc and minor arc, we used two pair of primers, F: 5′ TGGCTCCTTTAACCTCTCCA 3′; R: 5′AGGCTAAGCGTTTTGAGCTG 3′ and F: 5′ CTCCTCAAAGCAATACACTG 3′; and R: 5′ AAGGATTATGGATGCGGTTG 3′, respectively. The amplified fragments of the 77 samples and their corresponding controls were analyzed with the fragment analyzer (5200 Fragment Analyzer System, Agilent, Santa Clara, CA, USA).

#### 2.3.3. Evaluation of 6mA Methylation Sites

Overall, 6mA (6-methyl adenine) is a more common nucleotide modification than 5mA (5-methyl cytosine) in the human mitochondrial genome [34]. The methylation status of 6mA in the mitochondrial DNA in specific motifs was evaluated by digestion with restriction enzymes. DpnI (New England Biolabs, R0176) is sensitive to 6mA methylation and digests only methylated 5′-GATC-3′ sequences, and MboI (New England Biolabs, R0147) digests only unmethylated 5′-GATC-3′ sequence. We then applied the restriction enzyme digestion assay, followed by semi-quantitative PCR, to evaluate the methylation status on specific motif sequences (CTC/AATC) [35]. We investigated 3 sites with 6mA (6mA-1 for *Dloop*, 6mA-2 for *ND2*, and 6mA-6 for *ND6*) and one site without 6mA (6mA-N2 for *Cyb*), as identified previously by Hao et al. [36].

### 2.4. Sequencing of Nuclear Genes Involved in Mitochondrial DNA Integrity

The coding regions of *POLG1* (NM_002693), *C10ORF2* (Twinkle) (NM_021830), and *TFAM* (NM_003201) genes were sequenced with the dideoxy nucleotide termination method (Sanger sequencing). PCR was performed on cDNA, using primers that amplify the complete coding region of the transcript. All identified cDNA variants were confirmed on the corresponding genomic DNA. For the analysis of *DGUOK* (NM_080916) and MPV17 (NM_002437), all exons were amplified with intronic primers and sequenced with the Sanger methodology on an ABI 3500 sequencer (Applied Biosystems) (Appendix D) (see Appendix A for primers).

### 2.5. Transcriptome Analysis

#### 2.5.1. Microarray Gene Expression Analysis

Samples used for microarray analysis had RNA integrity number (RIN) values ≥ 8.5. Thirty-nine samples were excluded from this analysis, due to the lack of samples or to low RIN values. Generated sense-strand cDNA, using the Ambion WT Expression Kit (Life Technologies, Carlsbad, CA, USA), was fragmented, labeled, and hybridized to Human Gene 1.0 ST arrays (Affymetrix, Santa Clara, CA, USA), according to the manufacturer’s protocols. The arrays were washed, stained, and scanned in a GeneChip Scanner 3000 7G (Affymetrix, Santa Clara, CA, USA). CEL files were analyzed with Transcriptome Analysis Console (TAC) Software V4.0 (Affymetrix). Genes with *p*-values < 0.05, FDR-corrected *p*-values < 0.25, and fold change <−2 or >2 were considered significantly regulated genes.

Since we were analyzing tumors from two different populations, we are aware of the possibility of different handling issues, even though the sample handling was performed carefully and similarly. The gene-expression patterns were analyzed separately by performing Gene Set Enrichment Analysis (GSEA), using GSEA software V4.0.3 (Broad institute) (accessed on 27 September 2021) and referring to the Molecular Signatures database (MSigDB) for the “C2: Canonical pathways” gene sets.

#### 2.5.2. Reverse Transcriptase-qPCR (RT-qPCR)

Complementary DNA (1 ng per sample) served as a template for quantitative PCR, with the use of a QX200 ddPCR system (Bio-Rad, Hercules, CA, USA) and specific primers for 2 candidate genes, namely *MT-ND2* (assay ID: dHsaCPE5043508, Bio-Rad, CA, USA) and *MT-ND6* (assay ID: dHsaCPE5043480, Bio-Rad), labeled with FAM dye and one reference gene, namely *GUSB* (Beta Glucuronidase) (assay ID: Hs99999908_m1, Thermo-Fisher, USA), labeled with VIC. The gene-expression data were processed by using Quanta Soft software V1.7.4 (Bio-Rad, Hercules, CA, USA). Each reaction was performed in duplicate.

Given their involvement in mitochondrial biogenesis and autophagy, *P62*, *LC3*, *PCG1α*, and *NRF1* were explored. *GUSB* was used as an endogenous control gene. RT-PCR analysis was performed by using SYBR Green, with which we used 10 ng of cDNA; all reactions were performed in duplicates, using the 7500 fast real-time PCR system (Applied Biosystem, Foster City, CA, USA) (see Appendix A for primers).

### 2.6. Mass Spectrometry and Protein Analysis

The protein samples (mitochondria) were reduced by 5 mM of ditiothreitol, alkylated with 10 mM iodoacetamide, and incubated at 37 °C overnight with mass-spectrometry-grade trypsin (Thermo-Fisher, USA). The obtained peptides were desalted on Pierce™ C18 Tips (Thermo-Fischer, USA), according to manufacturer’s instruction. The desalted peptides were vacuum-dried, diluted with 12 µL 0.1% formic acid, and used for mass spectrometry analysis (see Appendix E for further information). Protein-enrichment analysis to identify related pathways and their functional connectivity was performed by using ProtExA (http://bioinformatics.cing.ac.cy/protexa/) (accessed on 27 October 2021), a web tool for the statistical and functional analysis of protein-expression datasets [37].

### 2.7. Statistical Analysis

Generation of Receiver Operating characteristic (ROC) curve in SPSS statistical package (Version 25.0), with the area under curve (AUC), cutoff, sensitivity, and 1-specificity values, served to set mtDNA content cutoff value, discriminating between normal and tumor tissue (Appendix E).

For genes and proteins expression, the quantitative differences were statistically analyzed by using Student’s *t*-test; differences with *p*-values lower than 0.05 were considered statistically significant. Graphs were generated by using GraphPad Prism software V5.0.

## 3. Results

### 3.1. Novel Mutations in PCCs/PGLs Susceptibility Genes

To examine mutations in these two cohorts, we designed a gene panel comprising all known and PCCs/PGLs-implicated susceptibility genes. In this design, 26 genes were covered by 706 probes. Of these, 99.8% of the enriched regions yielded sequence reads with a mean depth of 1000× per target and sample. All 41 identified mutations were confirmed with Sanger sequencing, and their status was checked in corresponding normal tissues (blood for all Swedish samples or non-tumor adrenal medullas for all French samples). Our analysis of paired normal DNA revealed that 25 mutations were somatic (61%), 10 were germline (24.4%), and six could not be assessed due to the lack of normal tissue (14.6%). Bioinformatic analysis suggested that 41 tumors presented mutations which are most likely the drivers, with seven mutations being novel. Three mutations in the *NF1*; a deletion of 12-bp in exon 27 and two frameshift mutations, c.5704_5705insC (L1902fs) and c.1904_1907 del (P635fs); two mutations in *EGLN1*, a frameshift mutation c.607_619 del(N203fs), and a non-sense mutation c.153G>A (W51X). Two tumors displayed frameshift and nonsense mutations in *SDHD* (c.205_217 del (E69fs)) and *CSDE1* (c.1660C>T (R554X)), respectively. In total, 53.2% of all PCCs/PGLs had mutations in the targeted susceptibility genes, leaving many tumors without a genetic explanation (Table 1) (Figure 1A).

### 3.2. Molecular Profiling and Gene Expression Patterns

Unsupervised hierarchical clustering based on Pheo-Type gene-expression profiling developed by Flynn et al. (2016) was performed and divided our cohort of PCCs/PGLs into five signatures: RTK1-3 (*RET, NF1, TMEM127*, and *HRAS*), *VHL*, and *SDHx* [6] (Figure 2).

After classifying the samples according to mRNA expression subtypes, we added gene-expression data for the mtOXPHOS genes (Figure 1). Most of the mtOXPHOS genes are generally upregulated over all Pheo-Type subgroups, except for MT-ND2/ND6 genes, which are downregulated, indicating probably a mitochondrial impairment in complex I among all PCCs/PGLs.

Our analysis of gene expression patterns revealed a difference and separated the Swedish and French cohorts into two groups. To better infer the biological differences, the two cohorts were analyzed separately, and a gene-set enrichment analysis was performed. GSEA determined the functional categories of significantly affected genes and yielded 14 gene sets that are enriched in the Swedish cohort: 12 gene sets are downregulated and associated mainly with genes involved in protein metabolism and RNA translation, and two upregulated gene sets associated with GPCR (G protein coupled receptors) signaling (Appendix A).

### 3.3. Mitochondrial DNA Alterations in PCCs/PGLs

#### 3.3.1. Novel Mitochondrial Mutations

Massive parallel sequencing of mtDNA revealed several variations in both known and novel polymorphic sites and allowed us to classify the 77 PCCs/PGLs patients into different haplogroups with different distributions separating the French and Swedish cohorts, but no significant association was found between mtDNA haplogroup bins and PCCs/PGLs (Appendix A).

All PCCs/PGLs carried mtDNA variants in the coding and non-coding regions of the mt genome. Among them, 10 were reported as pathogenic mutations in the MITOMAP database (https://www.mitomap.org/foswiki/bin/view/MITOMAP/WebHome) (accessed on 23 September 2021) and associated with cancer, and 21 were novel synonymous (1), missense (13), nonsense (2), frameshift (1), promotor (1), tRNA (2), or rRNA (1) mutations (Table 2). By using MitImpact2, PON-mt-tRNA, and mFold prediction programs, 12 variants were classified as pathogenic (seven somatic, four germline, and one unknown), and nine variants were considered polymorphisms (Table 2 and Figure 1B). Of the latter, 13 cases harbored complementary mutations (16.9%) to the nuclear susceptibility mutations (53.2%) (Figure 1A).

The POTTER software was used to predict the effect of the novel two nonsense and frameshift somatic mutations on the secondary structures of the ND1, ND6 (complex I, NADH Coenzyme Q oxidoreductase), and COXIII (complex IV, Cytochrome c Oxidase) proteins, respectively (Appendix A). The screening of genetic variants in mitochondrial genes revealed eight novel missense mutations, including three germline mutations (m.8466A>T, m.12068A>G, and m.15471T>C) in *ATP8*, *ND4*, and *Cyb* genes, respectively; four somatic mutations (m.4146A>C, m.4789G>A, m.13345G>A, and m.13498G>A) in *ND1*, *ND2*, and *ND5* genes, respectively; and one unknown mutation (m.9349T>C) in the *COXIII* gene, because neither blood nor normal adrenal medulla tissue was available (Figure 2). These variants are novel and not reported in the MITOMAP database, and in silico analysis predicted them as damaging (Table 2).

Furthermore, among the two novel mutations in tRNA genes, only the m.5658T>C identified in *MT-TN* (*tRNAAsn*) was predicted to be pathogenic with the PON-mt-tRNA program (Table 2). Indeed, this m.5658T>C change is located in position 72 of the acceptor stem, which changes the A:U Watson–Crick base pair to a G:U wobble pair. Moreover, the alignment of tRNAAsn sequences from different species showed that the A nucleotide is conserved through different species (Appendix A).

#### 3.3.2. Mitochondrial DNA Depletion and Large Deletions

The mtDNA contents of the 77 PCCs/PGLs and normal adrenal medulla tissues (only available from the French cohort) were measured by ddPCR. A reduction of copy number with more than 50.7% mtDNA content was considered as a depletion. A cutoff value distinguishing between tumors and controls was identified by using ROC curve with 93.8% sensitivity and 89% specificity (Figure 3A). We found that only 10 tumors (five from the French cohort (FR3, FR30, FR33, F36, and F39) and five from the Swedish cohort (PH40, PH57, PH64, PH65, and PH68)) presented a similar copy number as the normal adrenal medulla tissues. However, the mean mtDNA content of the rest of the tumors was nearly four times lower (3.8-fold) than that in normal PCCs/PGLs tissues (*p* < 0.0001) (Figure 3B) (Appendix A).

To assess the mtDNA integrity and mitochondrial rearrangements, we amplified a long fragment (~14 Kbp), including the major arc of mtDNA of the 77 samples and their corresponding controls (Appendix A). The result from the fragment analysis showed that, whilst control samples produced a detectable band of the expected size, 26% (20/77) of PCCs/PGLs presented heteroplasmic somatic deletions with different lengths (Figure 3C). These deletions are found in the major arc of mtDNA (between the origins of heavy and light strand replication sites), but not in the minor arc (data not shown). The deletions result in the elimination of several genes located in this region, such as tRNA genes and genes encoding respiratory chain proteins. By the amplification of the *MT-ND6* gene, we noticed the presence of a heteroplasmic somatic deletion (300 bp) in 39.1% (9/23) of the Swedish tumors, but not in the French cohort or in the controls (Figure 3D).

#### 3.3.3. Dysregulation of Mitochondrial Genes and Proteins Expression

Gene-expression analysis of the mitochondrial OXPHOS genes showed that there were no differences in mtRNA levels in the tumors presenting novel mitochondrial mutations, as compared to the rest of PCCs/PGLs, in the corresponding mutated genes. On the other hand, analyzed tumors from both the Swedish and French cohorts presented a lower MT-ND2 and MT-ND6 gene expression than normal tissues (*p* < 0.0001) (Figure 4A). In addition, the Swedish cohort had a significantly lower gene expression of MT-ND6 when compared to the French cohort (*p* < 0.05).

Mass spectrometry analysis of isolated mitochondrial proteins showed a slight increase of expression levels of mtOXPHOS proteins (ND4, ND5, COXII, ATP6, and ATP8) in the tumors, presenting different variations in comparison to normal adrenal medulla tissues (*p* > 0.05) (Table 3).

A set of protein-expression data across samples were analyzed by using the ProtExAc program. After performing statistical analysis and filtering, we sorted out the differentially expressed proteins and performed enrichment analysis to identify top-scored pathways. A total of 253 proteins were further clustered into 117 groups basis on their enrichment score. The pathway-enrichment analysis showed that most of the proteins were involved in phagosome, oxidative phosphorylation, TCA cycle, pyruvate metabolism, and glutathione metabolism, in addition to other cell-signaling pathways (Figure 4B).

#### 3.3.4. Absence of 6mA Methylation

Our strategy was to check whether 6mA at the promotor region of both*ND2* and *ND6* genes could affect the mitochondrial transcription and the low mtDNA copy number. We investigated three sites within previously identified 6mA (Dloop, ND2, and ND6). The 6mA restriction enzyme and semi quantitative PCR identified none of these sites to be methylated (Appendix A).

### 3.4. Absence of Mutations in Mitochondrial DNA Integrity Genes

The identification of mtDNA alterations (mtDNA deletions/mtDNA depletion) may also be due to Mendelian disorder, since the maintenance of mtDNA integrity is dependent on several nuclear-encoded proteins. In this context, our microarray data and recent studies (van Gisbergen et al., 2015; Sharma and Sampath, 2019) were used to select the susceptibility nuclear genes, *POLG1*, *C10ORF2*, *DGUOK*, *TFAM*, and *MPV17*, which are critically involved in maintaining mtDNA integrity for mutation and gene-expression analysis in our cohorts. No pathogenic mutations were identified, but three common polymorphisms were detected: rs3087374 and rs2307441 in *POLG1* in eight and two tumors, respectively; and rs74874677 in *DGUOK* in two different tumors.

### 3.5. Dysregulation of Mitochondrial Biogenesis and Mitophagy

To assess mitochondrial biogenesis, *PCG1α*, *NRF1*, and *TFAM* gene expression was investigated. Tumors had a very low *PCG1α*, *NRF1*, and *TFAM* gene expression (*p* = 0.02; *p* < 0.001 and *p* < 0.001, respectively) as compared to normal tissues (Figure 5A). In addition, we investigated two mitophagy markers and we showed a higher expression of *LC3a* (*p* = 0.0014) and a lower expression of *P62* (*p* < 0.0001) in those tumors (Figure 5B).

## 4. Discussion

In the present study, we designed a targeted NGS assay for mutation analysis of 26 PCCs/PGLs susceptibility genes, followed by the mRNA expression analysis, using a gene list that distinguishes between pseudohypoxic and RTK/RAS-driven PCCs/PGLs and mtOXPHOS. In general, we observed that the mtOXPHOS mRNA levels were upregulated across the Pheo-Type-based gene-expression signatures, with downregulation of *MT-ND2/ND6* genes, encoding two subunits of complex I. The differences in complex I gene expression have been observed previously in many cancers (e.g., lung, colon, and bladder) [41,42]. Possible explanations are post-transcriptional regulatory mechanisms and/or differences in mRNA stability among mtDNA-encoded genes. Additionally, nuclear gene expression and transcriptional elements of OXPHOS subunits and assembly factors could be factors for the observed changes in the transcript levels [41,42,43]. The regulation of complex I subunit mRNA is more complex, rather than being a global co-regulation [43,44], indicating probably an impairment of the OXPHOS complex I. This observation is further substantiated by protein-enrichment analysis, where the OXPHOS pathway is on the top of the enriched KEGG pathways.

On the other hand, the observed transcriptomic differences between the Swedish and French tumors may be explained by the different geographic (ethnic) origins, as was previously observed between Scandinavian and French PCCs/PGLs during a genetic analysis of SNPs and PCC/PGL susceptibility [25]. The mitochondrial haplogroup analysis confirmed these population differences (Appendix A).

In addition, we performed sequencing of the entire mitochondrial DNA from 77 PCCs/PGLs patients. We identified eight known pathogenic mutations previously associated with cancers and 21 novel mutations, where 12 are considered pathogenic. Over the past decades, somatic and germline alterations in nuclear genes and epigenetic changes have been analyzed to identify the molecular basis for tumor formation. Recently, changes in mitochondrial cellular content and mtDNA mutations have been proposed as new molecular markers for cancer detection and surveillance [45]. The frequency of mitochondrial mutations in solid tumors is less studied, including bladder and head and neck cancer where hot-spot genes for mtDNA mutations are identified: *ND3, ND4, COXI/II/III, 16S rRNA*, and deletions in the Dloop region [46,47].

Frameshift mtDNA mutations are rare and were previously reported in six mitochondrial genes: *MT-CYB, MT-ND1//4/5/6*, and *MT-COXIII*, with only two frameshift mutations (14510delA and 13384insC) reported in the *MT-ND6* gene. Previous studies reported that *MT-ND6* frameshift mutations in cell lines manifest a complex I enzymatic deficiency and reduced levels of the assembled complex I, suggesting increased cell death and disease pathogenesis [48]. Using in silico prediction and referring to the functional studies in MT-ND6 gene, we see that the novel 14603insT (Ser24Tyrfs) mutation is considered as pathogenic as the two cited mutations. Besides, screening the *MT-NDx* genes displayed six somatic and one germline pathogenic mutations, including a nonsense somatic mutation in *MT-ND1* gene m.3563G>A (Trp86X). Although the microarray and mass spectrometry data failed to confirm expression alterations, the mutations are predicted to disturb complex I formation and formation of the proton translocation module (P module), as indicated by the Mitomap2 database. Previously, mutations in *MT-NDx* genes have been associated with different human cancers (e.g., colorectal carcinomas, pancreatic cancer, and oral squamous cell carcinoma) [49]. The *MT-NDx* mutations inhibit oxidative phosphorylation, increase ROS production, and potentially stimulate the metastatic potential in myeloid leukemia and contribute to carcinogenesis [50].

The screening of *MT-COX* genes revealed one novel somatic heteroplasmic mutation in *MT-COXIII* m.9553G>A (Trp116X). This pathogenic mutation results in a truncated protein with the loss of 145aa of the COX3 polypeptide. Mutations in mitochondrial genes encoding MT-COXI/II/III subunits (complex IV) have been associated with numerous cancers. *COXIII* mutations have mainly been associated with colon and thyroid cancer (https://www.mitomap.org/foswiki/bin/view/MITOMAP/MutationsSomatic) (accessed on 23 September 2021) and suggested to cause an overproduction of superoxide anions by the mitochondrial respiratory chain at COX deficiency [51].

Pathogenicity of the m.5658T>C germline mutation in the tRNAAsn gene has been assessed, suggesting the absence of essential recognition nucleotides. A previous study of the steady-state aminoacylation kinetics of acceptor stem mutant tRNAs demonstrated that replacing any base pair in the acceptor helix with another Watson–Crick base pair affects aminoacylation (or tRNA “charging”) efficiency and can elicit mistranslation or lead to a defective tRNA [52]. 

Certain mutations in mitochondrial DNA produce oxidative phosphorylation defects (preferentially in complex I) that may have pro-tumorigenic effects [53]. For example, MT-ND6 mutations lead to deficient complex I activity and high ROS generation, making the cells highly metastatic. Ishikawa et al., found that replacement of endogenous mtDNA in a poorly metastatic mouse tumor cell line with mtDNA from a highly metastatic cell line (Lewis lung carcinoma) containing two pathogenic mutations in *MT-ND6* made the recipient tumor cell acquire the metastatic potential of the transferred mtDNA. Furthermore, these mitochondrial mutations lead to a deficiency in respiratory complex I, associated with overproduction of ROS. Pretreatment of the highly metastatic tumor cells with ROS scavengers suppressed their metastatic potential in mice [54].

The mitochondrial deletions identified in our study could be one of the mitochondrial mutations identified to be causative of human diseases and one among 200 documented pathogenic mtDNA mutations (http://www.mitomap.org/) (accessed on 23 September 2021) [55]. Single and large-scale mtDNA deletions are sporadic and occur because of abnormalities in mtDNA replication and can also arise during the repair of mtDNA damage [56,57]. Large deletions, observed in 16/23 of Swedish tumors and 4/54 of French tumors, result in the loss of several protein-encoding genes and some tRNA genes. Moreover, a deletion of 300 bp was detected only in the Swedish tumors and may have a tumorigenic role similar to the common 4977 bp deletion in the major arc, which has the potential to be a biomarker for cancer occurrence in the tissue [56]. This is interesting, but due to a small sample size, this finding needs to be confirmed in larger PCC/PGL cohorts. Thus, they may affect the mitochondrial polypeptide synthesis and contribute to the impairment of the oxidative phosphorylation and energy metabolism in the respiratory chain [58]. This finding, is in accordance with Neuhaus et al. (2014, 2017) observations, where they have showed that catecholamine metabolism could induce mtDNA deletions, causing mitochondrial dysfunction in adrenal medulla and cortex [15,16]

Another alteration that may affect cellular function is the copy number and depletion of mtDNA that are observed in PCCs/PGLs. Changes in the tumor mtDNA copy number have been noted in human cancers [59] and found to vary between different cancer forms, being elevated in, for example, primary tumors of head and neck squamous cell carcinoma [60] and in papillary thyroid carcinomas [61], and reduced in breast tumors [62] and gastric cancers [63], relative to normal controls. It has been demonstrated that the reduction of mitochondria leads to tumorigenesis by inducing changes in redox status, membrane potential, ATP levels, gene expression, nucleotide pools, and increased chromosomal instability [64]. Still, it is not yet known whether the depletion of mtDNA is a consequence of losses of mitochondrial function typical of cancer cells, or if it leads to a mutant phenotype adapted to the needs of the cancer cell [65].Further studies are necessary to clarify these issues. On the other hand, our data showed a significant decrease of *PCG1α*, *NRF1*, and *TFAM* gene expression in the analyzed tumors compared to normal tissues. PGC1α is a central regulator of certain nucleus-encoded mitochondrial genes through interactions with multiple transcription factors, including nuclear respiratory factors such as NRF1. NRF1 binds to the promoter and regulate the expression of TFAM and mitochondrial matrix proteins (involved in the replication and transcription of mtDNA) [66,67]. By activating TFAM, PGC-1a could regulate mitochondrial biogenesis, including an increase of mitochondrial copy numbers, activation of the respiratory chain, and augmentation of OXPHOS capacity [68,69]. Our results showed that the reduced number of mitochondria was accompanied by a decrease of the mitochondrial biogenesis, suggested in many recent studies to have a crucial role in the decrease in tumorigenesis or tumor progression [70,71].

Mitochondrial biogenesis, dynamics (fusion and fission), and autophagy (mitophagy) control content and quality of the mitochondria [72]. The impairment of these mechanisms has been associated to cancer, i.e., mitochondrial dynamics [73]. Indeed, DRP1 activation increases the mitochondrial fission in several types of cancers [74] and promotes mitochondrial fragmentation facilitating cancer-cell migration and invasion [75]. Here, we showed a significant overexpression of DRP1 protein (Dynamin-1-like protein) in the analyzed tumors (Table 3), accompanied by decreased mitochondrial biogenesis markers in PCCs/PGLs and a reduced number of mtDNA, which are indicative of mitophagy. To monitor mitophagy, we have extended the expression information on additional biomarkers. Our gene-expression data showed a significant upregulation of *LC3a* and downregulation of *p62*. Among the autophagy-related proteins, LC3a is involved in autophagosome formation, and p62 serves as a selective autophagy substrate; both are widely used autophagy markers for monitoring autophagy activity. The complex regulation of LC3a and p62 is suggested to be related to autophagy activity in cancers [76]. These results could suggest the high level of mitophagy [77,78]. We believe that PCCs/PGLs tumor cells may adopt an overactive process of mitochondrial turnover, leading to the selective elimination of dysfunctional mitochondria through mitophagy [79].

## 5. Conclusions

Taken together, our results showed that the PCCs/PGLs are caused by germline or somatic mutations in known susceptibility genes in about 50%, and complementary mutations in mitochondrial genes are present in about 17%. These tumors harbored different mitochondrial mutations, deletions, and a reduced copy number and displayed different gene expression patterns, several of which appear without any identified mutation in susceptibility genes, indicating a complementarity and a potentially contributing role in the tumorigenesis PCCs/PGLs.

## Figures and Tables

**Figure 1 cancers-14-00269-f001:**
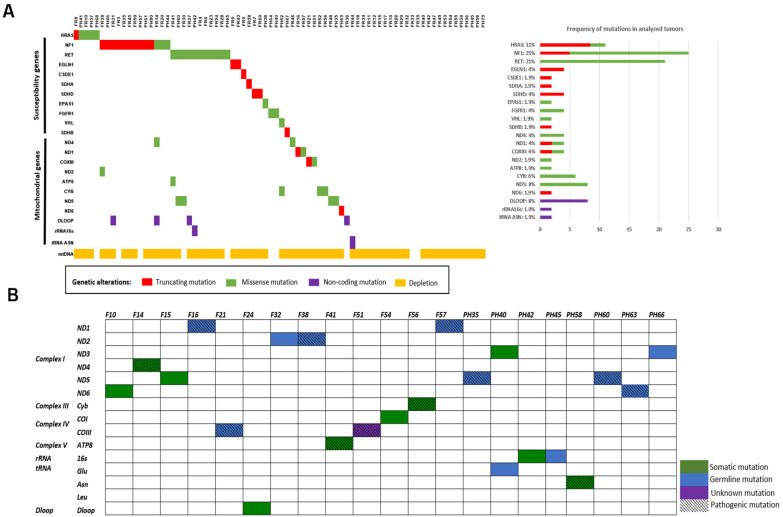
Mutation’s analysis. (**A**) OncoPrint summarizing the distribution of nuclear and mitochondrial pathogenic mutations, and mitochondrial depletion in PCC/PGL tumors with total mutation frequencies. The OncoPrint showed an overview of genomic alterations (legend) in particular genes (rows) affecting individual samples (columns). The truncated, missense, and non-coding mutations are shown in red, green, and purple, respectively. Mitochondrial depletion is shown in yellow. (**B**) Summary of 21 novel gene mutations detected in 20 cases of PCCs/PGLs from France and Sweden; germline mutations are denoted by green color; somatic mutations are denoted by blue color, unknown mutations are denoted by purple color, and pathogenic mutations are denoted by hatch mark.

**Figure 2 cancers-14-00269-f002:**
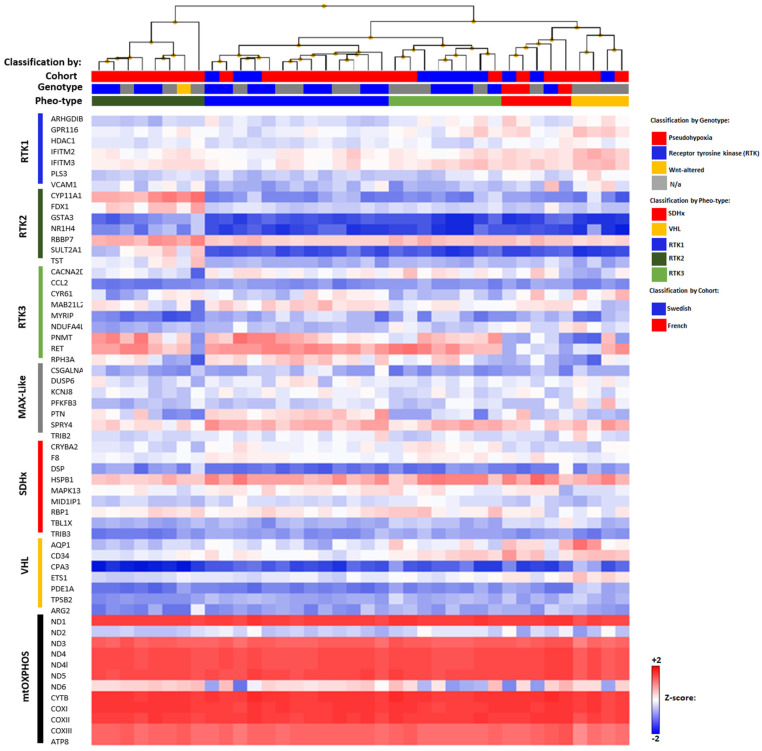
Gene-expression heatmap of the Pheo-Type gene set (RTK1-3, MAX-like, VHL, and SDHx) and the last gene set displays the mtOXPHOS genes in Swedish and French PCCs/PGLs with known genotype. Genotypic subtype indicated as pseudohypoxic (red: *VHL*, *SDHx*, *EPAS1*, and *EGLN1* mutations), receptor tyrosine kinase (RTK)-driven (blue: *NF1*, *HRAS*, *RET*, *FGFR1*, and *TMEM127* mutations), WNT-altered (yellow; *CSDE1* mutations), and not applicable (N/a, gray; no mutation). Pheo-Type cluster classification indicated as pseudohypoxic (*SDHx*, red; *VHL*, yellow), RTK1 (blue), RTK2 (green), and RTK3 (clear green).

**Figure 3 cancers-14-00269-f003:**
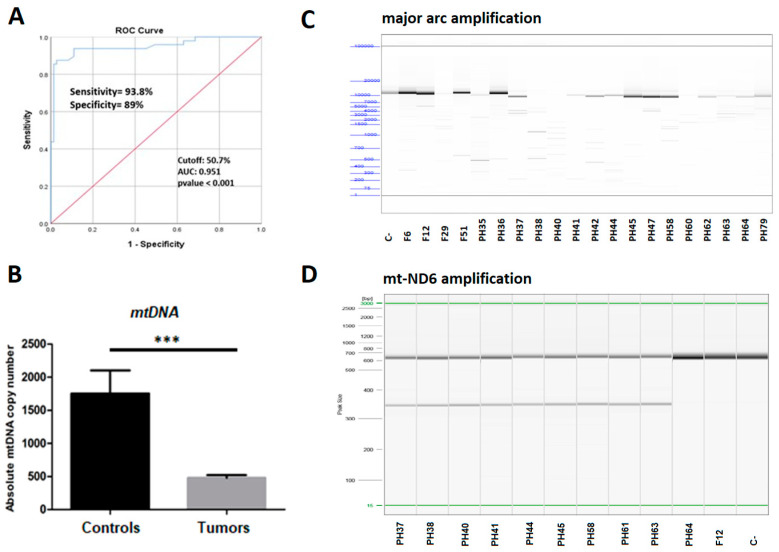
Mitochondrial DNA depletion and large deletions analysis. (**A**) ROC curve to assess the optimal cutoff value of mitochondrial DNA copy number (51%). AUC, area under the curve; ROC, receiver operating characteristic. (**B**) Absolute mtDNA copy number in the PCCs/PGLs (Swedish and French cohorts) and controls (normal adrenal medulla tissues); *p*-values were calculated by *t*-test, *** <0.0001. (**C**) Large mitochondrial DNA deletions detected in PCCs/PGLs from the Swedish and the French cohort, using the Fragment Analyzer. (**D**) Gel electrophoresis separation of a 300 bp heteroplasmic deletion detected in *MT-ND6* gene, using QIAxcel (Qiagen) in 9 tumors from the Swedish cohort (right). (C-: negative control yielded the expected band size.)

**Figure 4 cancers-14-00269-f004:**
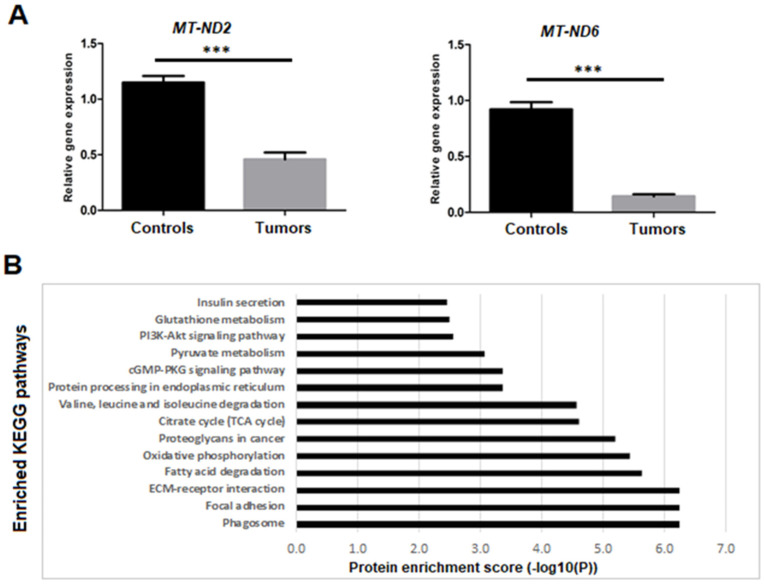
Mitochondrial genes and proteins expression analysis. (**A**) Relative expression of MT-ND2 and MT-ND6 with significant difference between tumors and controls (*p*-values calculated by *t*-test, *** <0.0001). (**B**) Functional enrichment analysis of differentially expressed proteins, using PROTEXA program. Bar graph of enriched KEGG biological pathways in proteins is shown.

**Figure 5 cancers-14-00269-f005:**
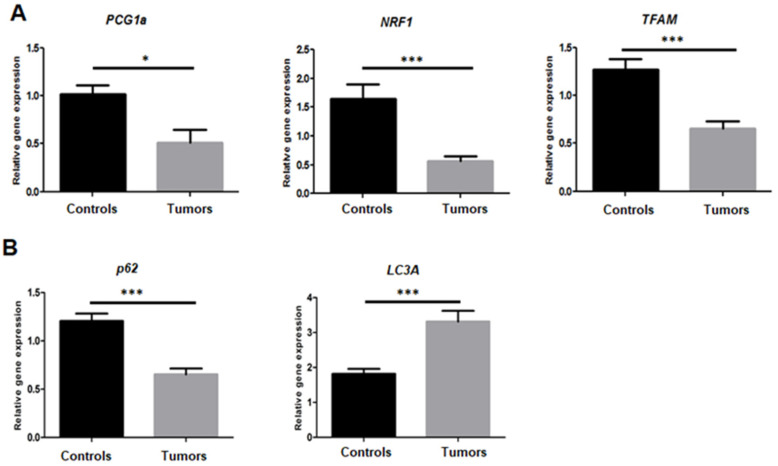
Mitochondrial biogenesis and mitophagy analysis. (**A**) Relative expression of mitochondrial biogenesis markers: PCG1α, NRF1, and TFAM genes with significant difference between tumors and controls (*p*-values calculated by *t*-test; * <0.05 and *** <0.0001). (**B**) Relative expression of mitochondrial mitophagy markers: P62 and LC3a, with significant difference between tumors and controls (*p*-values calculated by *t*-test; *** <0.0001).

**Table 1 cancers-14-00269-t001:** Pathogenic mutations identified among 41 patients with pheochromocytomas and paragangliomas from Nancy, France, and from Linköping, Sweden.

Sample	Gene	Mutation	Protein Variation	Mutation Status	Previously Reported
** *French cohort* **					
F3	*NF1*	c.2044C>T	Q682X	Unknown	Yes (rs1597712392)
F4	*RET*	c.2753T>C	M918T	Somatic	Yes (rs74799832)
F5	*CSDE1*	c.1660C>T	R554X	Unknown	No
F6	*RET*	c.1900T>C	C634R	Germline	Yes (rs75076352)
F7 ^†^	*SDHD*	c.205_217 del	S69fs	Unknown	No
F8	*HRAS*	c.37G>C	G13R	Germline	Yes (rs104894228)
F9	*EGLN1*	c.153G>A	W51X	Unknown	No
F10	*HRAS*	c.181C>A	Q61K	Somatic	Yes (rs28933406)
F14	*NF1*	c.1885G>A	G629R	Germline	Yes (rs199474738)
F20	*NF1*	c.1466A>G	Y489C	Somatic	Yes (rs137854557)
F23	*RET*	c.1853G>A	C618Y	Somatic	Yes(rs79781594)
F24	*RET*	c.1900T>C	C634R	Germline	Yes (rs75076352)
F27 ^†^	*HRAS*	c.182A>T	Q61L	Somatic	Yes (rs121913233)
F28	*RET*	c.2753T>C	M918T	Somatic	Yes (rs74799832)
F30	*RET*	c.1832G>A	C611Y	Germline	Yes (rs377767397)
F31	*NF1*	c.5704_5705 insC	L1902fs	Somatic	No
F35	*NF1*	c.3674_3688 del	A1226_V1230 del	Somatic	No
F37	*RET*	c.2753T>C	M918T	Somatic	Yes (rs74799832)
F38	*NF1*	c.7301_7302 delAG	Q2434fs	Somatic	No
F41	*RET*	c.1902C>G	C634W	Somatic	Yes (rs77709286)
F43 ^†^	*EGLN1*	c.607_619 del	N203fs	Germline	No
F45	*NF1*	c.1904_1907 del	P635fs	Germline	No
F58	*SDHA*	c.1432_1432+1delGG	428?	Germline	Yes (rs878854627)
F59	*NF1*	c.6841G>T	G2281X	Somatic	Yes [38]
F60	*SDHD*	c.187_188 delTC	L64fs	Unknown	Yes (rs387906358)
** *Swedish cohort* **					
PH 37	*HRAS*	c.182A>G	Q61R	Somatic	Yes (rs121913233)
PH 38	*EPAS1*	c.1235T>A	I412N	Somatic	Yes [39]
PH 40	*FGFR1*	c.1638C>T	R546K	Somatic	Yes [25]
PH 41	*HRAS*	c.37G>C	G13R	Somatic	Yes [40]
PH 42	*RET*	c.1893_1898delCGAGCT	Asp631_Leu633delinsGlu	Somatic	Yes (rs121913307)
PH 44	*NF1*	c.1340T>C	L447P	Somatic	Yes [22]
PH 57	*NF1*	c.4798_4799delAA	K1600fs	Somatic	Yes [25]
PH 60	*RET*	c.2753T>C	M918T	Somatic	Yes (rs74799832)
PH 61	*NF1*	c.2806A>T	K936X	Somatic	Yes [25]
PH 62	*VHL*	c.284C>G	P95R	Somatic	Yes [25]
PH 64	*FGFR1*	c.1638C>T	R546K	Somatic	Yes (rs779707422)
PH 65	*RET*	c.2753T>C	M918T	Germline	Yes (rs74799832)
PH 66	*NF1*	c.289C>T	Q97X	Somatic	Yes (rs1597635615)
PH 67	*SDHB*	c.664delT	G228fs	Germline	Yes [25]
PH 68	*HRAS*	c.37G>C	G13R	Somatic	Yes (rs104894228)
PH80	*NF1*	c.3158C>G	S1053X	Unknown	Yes (rs1597717610)

^†^ Paragangliomas; unknown, no normal tissue available.

**Table 2 cancers-14-00269-t002:** Summary of the novel mitochondrial variants and known pathogenic mitochondrial mutations associated with cancer and other mitochondrial alterations detected in PCCs/PGLs from the Swedish and the French cohort.

	Gene	Mutation	Protein Variation	HM/HT State (%)	Mutation Status	Predicted Mutation Impact			mt CNV (%)	mt Large Deletion	Phenotype
						MitImpact2 ^a^	mfold ^b^	PON-mt-tRNA ^c^			
** *French cohort* **											
F10	*ND6*	14173T>C	Val167=	HM	Germline	NA	NA	NA	41.6	No	**Novel**
F14	*ND4*	12068A>G	Met437Val	HM	Germline	Deleterious	NA	NA	18.6	No	**Novel** **^†^**
	*Dloop*	16093T>C	non coding	HT (60.3)	Germline	NA	NA	NA			Breast, Thyroid, and Prostate cancers
F15	*ND5*	14143A>G	Thr603Ala	HM	Germline	Neutral	NA	NA	40.9	No	**Novel**
F16	*ND1*	3563G>A	Trp86X *	HT (39)	Somatic	NA	NA	NA	29.6	No	**Novel** **^†^**
F21	*COXIII*	9553G>A	Trp116X *	HT (39.7)	Somatic	NA	NA	NA	20.8	No	**Novel** **^†^**
F24	*Dloop*	16076C>T	non coding	HT (42.2)	Germline	NA	NA	NA	35.9	No	**Novel**
F31	*Dloop*	16183delA	non coding	HT (56)	Somatic	NA	NA	NA	22.0	No	Colon cancer
F32	*ND2*	4725A>C	Met86Leu	HT (65.6)	Somatic	Neutral	NA	NA	17.1	No	**Novel**
F36	*Dloop*	16093T>C	non coding	HT (67)	Germline	NA	NA	NA	188.9	No	Breast, Thyroid, and Prostate cancers
F37	*Dloop*	16183delA	non coding	HM	Germline	NA	NA	NA	9.6	No	Colon cancer
F38	*ND2*	4789G>A	Gly107Glu	HT (74.2)	Somatic	Deleterious	NA	NA	15.0	No	**Novel** **^†^**
F41	*ATP8*	8466A>T	His34Leu	HM	Germline	Deleterious	NA	NA	19.5	No	**Novel** **^†^**
F44	*Dloop*	16218C>T	non coding	HM	Germline	NA	NA	NA	17.6	No	Ovarian and Prostate cancers
F46	*ND5*	13135G>A	Ala267Thr	HM	Germline	Deleterious	NA	NA	40.9	No	Cervical and head and neck cancers
F48	*ND5*	12338T>C	Met1Thr	HM	Germline	Deleterious	NA	NA	3.5	No	Colon cancer
F51	*COXIII*	9349T>C	Leu48Pro	HT (35.9)	Unknown	Deleterious	NA	NA	22.2	No	**Novel** **^†^**
F52	*CYB*	15789C>T	Thr348Ile	HM	Unknown	Deleterious	NA	NA	14.6	No	Breast cancer
F54	*COXI*	6072A>G	Ile57Val	HM	Germline	Neutral	NA	NA	11.8	No	**Novel**
F56	*Cyb*	15471T>C	Leu24Ser	HM	Germline	Deleterious	NA	NA	7.7	No	**Novel** **^†^**
F57	*ND1*	4164A>C	Met286Ile	HT (73.9)	Somatic	Deleterious	NA	NA	10.0	No	**Novel** **^†^**
** *Swedish cohort* **											
PH35	*ND5*	13498G>A	Gly388Ser	HT (35.3)	Somatic	Deleterious	NA	NA	29.0	Yes	**Novel** **^†^**
PH40	*ND3*	10113A>G	Ile19Val	HM	Germline	Neutral	NA	NA	57.9	Yes	**Novel**
	*tRNAGlu*	14721G>A	NA	HT (57.3)	Somatic	NA	NA	Neutral			**Novel**
PH42	*rRNA16s*	2222T>C	non coding	HM	Germline	NA	Deleterious	NA	36.5	Yes	Pancreatic cancer
PH45	*rRNA 16s*	1969G>A	NA	HT (36.6)	Somatic	NA	Neutral	NA	46.6	Yes	**Novel**
PH58	*tRNAAsn*	5658T>C	NA	HT (30.8)	Germline	NA	NA	Deleterious	32.7	Yes	**Novel** **^†^**
PH60	*ND5*	13345G>A	Ala337Thr	HT (88.7)	Somatic	Deleterious	NA	NA	28.3	Yes	**Novel** **^†^**
PH62	*CYB*	15672T>C	Met309Thr	HM	Germline	Deleterious	NA	NA	20.2	Yes	Breast and Thyroid cancers
PH63	*ND6*	14603GinsT	Ser24Tyrfsx11	HT (30.5)	Somatic	NA	NA	NA	35.3	Yes	**Novel** **^†^**
PH66	*ND3*	10068G>A	Ala4Thr	HT (33.3)	Somatic	Neutral	NA	NA	29.0	No	**Novel**

^a^ MitImpact2 (http://mitimpact.css-mendel.it/) (accessed on 24 September 2021) a collection of pre-computed pathogenicity predictions for all nucleotide changes that cause non-synonymous substitutions in human mitochondrial protein-coding genes (analyzed missense mutations/variations); ^b^ mfold (http://unafold.rna.albany.edu/?q=mfold/RNA-Folding-Form) (accessed on 24 September 2021) used for nucleic acid folding and hybridization prediction (analyzed rRNA variations); ^c^ PON-mt-tRNA (http://structure.bmc.lu.se/PON-mt-tRNA/) (accessed on 24 September 2021) classifies all possible single nucleotide substitutions in all human mt-tRNA (mitochondrial transfer RNA) based on evidence from several sources and used the data to develop a multi-factorial probability. * Highlighted variations are a frameshift and nonsense mutations. ^†^ New mutations predicted as pathogenic. PCCs, pheochromocytomas; PGLs, paragangliomas; mt, mitochondrial; HM/HT, homoplasmic/heteroplasmic; NA, not applicable; CNV, copy-number variation; unknown, no available normal tissue.

**Table 3 cancers-14-00269-t003:** Protein expressions’ change of mtOXPHOS and other relevant proteins in tumors compared to normal tissues.

Identified Proteins	Gene Name	Fold Change	*p*-Value
**mt-OXPHOS**			
NADH-ubiquinone oxidoreductase chain 4	*ND4*	1.8	0.3
NADH-ubiquinone oxidoreductase chain 5	*ND5*	1.6	0.71
Cytochrome c oxidase subunit 2	*COX2*	0.9	0.61
ATP synthase subunit	*ATP6*	1.2	0.52
ATP synthase protein 8	*ATP8*	*	0.12
**Other Mitochondrial Proteins**			
Dynamin-1-like protein	*DNM1L*	7.4	0.011
Protein-L-isoaspartate O-methyltransferase	*PCMT1*	5.9	0.0029
Voltage-dependent anion-selective channel protein 3	*VDAC3*	2.6	0.045
ATP synthase subunit gamma	*ATP5F1C*	1.9	0.028
ATP synthase subunit d, mitochondrial	*ATP5H*	1.9	0.019
ATP synthase subunit gamma	*ATP5F1C*	1.9	0.028
ATP synthase subunit d, mitochondrial	*ATP5H*	1.9	0.019
NADH dehydrogenase (ubiquinone) 1 alpha subcomplex subunit 13	*NDUFA13*	1.8	0.045
NADH dehydrogenase (ubiquinone) 1 alpha subcomplex subunit 5	*NDUFA5*	1.3	0.028
Cytochrome c oxidase subunit 7C	*COX7C*	0.7	0.023
Cytochrome b-c1 complex subunit 6	*UQCRH*	0.6	0.0033
Trifunctional enzyme subunit alpha	*HADHA*	0.6	0.0066
Isocitrate dehydrogenase (NAD) subunit alpha	*IDH3A*	0.4	0.00086
Phosphoglycerate kinase 1	*PGK1*	0.4	0.004
Electron transfer flavoprotein–ubiquinone oxidoreductase	*ETFDH*	0.3	0.0028
Glutathione S-transferase	*GSTM3*	0.2	0.045
Pyruvate carboxylase	*PC*	0.1	0.024
Succinate-CoA ligase (GDP-forming) subunit beta	*SUCLG2*	0.06	0.00049
Acetyl-CoA acetyltransferase	*ACAT1*	0.02	0.00023
NADPH: adrenodoxin oxidoreductase,	*FDXR*	0.004	<0.00010

* Detected only in tumors.

## Data Availability

The data presented in this study are available on request from the corresponding author.

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
