# Peer review of "Genetic Alterations in Mitochondrial DNA Are Complementary to Nuclear DNA Mutations in Pheochromocytomas"

_cancers, 2022, doi:10.3390/cancers14020269_

Round 1

Reviewer 1 Report

Tabebi and collaborators analyzed a panel of 26 nuclear genes and the entire mtDNA sequence, as well as mtDNA copy number, large mtDNA deletion, and gene/protein expression in 77 pheochromocytomas and paragangliomas (PCCs/PGLs). Authors observed that 53.2% of the tumors had a mutation in at least one of the nuclear genes and 16.9% had mitochondrial mutations. Moreover, large deletions were found in 26% of tumors and decreased mtDNA copy number, which was accompanied with a reduced expression of the regulators that promote mitochondrial biogenesis and altered expression of two mitophagy genes, occurred in more than 87% of tumors.

Overall the manuscript is well written and results are of interest. However, there are some concerns that should be addressed to improve the manuscript.

Main comments.

Materials and Methods.

A table with the main characteristics of the study population (e.g., sex, age, etc.) should be provided. More information regarding normal tissue should also be provided (e.g., what investigations were done to make sure there were no contamination of cancer cells? what is the distance from the tumor tissue?).

The rationale used to choose the 26 susceptibility genes should be more detailed.  In the “Abstract” authors stated that “To determine the potential roles of mtDNA alterations in sporadic PCCs/PGLs, we analyzed a panel of 26 nuclear susceptibility genes…”. Are all the 26 genes potentially involved in mitochondrial alterations?

Results.

How was the difference in mtDNA content between tumor and normal tissue considering only the French samples for which the two tissue types were available? This analysis should be added at least in supplementary material.

Given its importance in the manuscript, the figure of ROC curve should be moved in the main text.

Table 3 is missing.

Minor comments.

Name of the genes should be in italics throughout the manuscript.

Reviewer 2 Report

The paper is interesting on the analysis of mtDNA mutations, mtRNA, and protein expression that were not previously widely studied for pheochromocytomas and paragangliomas. However, the study include redundant data analysis, which purposes are not obvious, and the results obtained were poorly interpreted (e.g., comparison of gene expression in two cohorts studied). Moreover, the study was performed predominantly on pheochromocytomas so it is not correct to include paragangliomas in the title of the paper. Also, the manuscript is not well-written and contains biological and semantic errors. The manuscript requires thorough correction and revision.

In the section “Introduction” there are many mistakes in the biology of paragangliomas. For example, authors wrote that PCCs/PGLs are catecholamine-producing tumors but not all paragangliomas produce catecholamines. Most parasympathetic paragangliomas are non-functional.

“Probably more than 30% of pheochromocytomas are hereditary…” – About 40% of paragangliomas (PCCs and PGLs) are hereditary according to WOS.

“…germline mutations in well-known cancer susceptibility genes…” – What are the well-known cancer susceptibility genes in this context? Authors should list the genes.

“Many sporadic pheochromocytomas have somatic mutations in one of these hereditary genes…” – Somatic mutations in genes associated with hereditary paragangliomas are not often.

“The known germline and somatic mutations account for the pathogenesis of approximately 60% of all PCCs/PGLs..” – Are authors talking about known mutations or known genes?

“Cluster 1 tumors (mutations in e.g., VHL, SDHA/B/C/D/AF2 genes)…” – The list of genes related to Cluster 1 is incomplete. Moreover, there are known at least three clusters, not two (the third - Wnt signaling).  

The sentence “Pseudohypoxia clustering is defined by overexpression of…” – Is not clear, what the authors want to say. The Pseudohypoxia cluster does not include MAX-like tumors.

“mtDNA mutations and copy number alterations have been associated with high levels of reactive oxygen species (ROS, produced during oxidative phosphorylation (OXPHOS), and with a less efficient mtDNA repair system and oxidative mtDNA damage is believed to be associated with tumor malignancy” – The essence of this proposal is not clear. The sentences should be rephrased.

Two sentences “..but also in the mitochondrial genomes, which has not been explored before in PCCs/PGLs” and “…mitochondrial genome analysis is not investigated in PCCs/PGLs…” are about the same.

“…publications have shown that catecholamine metabolism is fundamental to mtDNA integrity and mitochondrial function” – It is not clear what authors want to say. As I mentioned above, not all paragangliomas produce catecholamines (e.g., head and neck PGLs) but HNPGLs often harbor SDHx mutations and are characterized by SDH complex deficiency.

“We investigated a panel of 26 susceptibility genes..” – I recommend correcting this as something like “we investigated mutational status/mutations in a panel of 26 susceptibility genes”.

It is very important to provide the clinic-pathologic characteristics of the studied tumors (age, gender, metastasis, recurrence, multifocality, etc.). What is the localization of 3 PGLs included in the study? Were tumor tissues fresh-frozen or FFPE? Were normal fresh-frozen tissues adjacent?

How was the panel of 26 sensitivity genes formed? This list is a mixture of genes, in which there are often germline mutations or one in which only somatic mutations have been found. I recommend presenting a list of classic paraganglioma susceptibility genes and other genes with links to studies that have described them with mutations in paragangliomas. In addition, not all-important genes have been included in the list, for example, there is no SLC25A11, hTERT, TP53, IDHx, MET, MEN1, and others. I recommend testing the mutational status of these genes.

Was sequencing on MiSeq performed at 151x2 mode?

The authors studied 77 patients but there is no information about how many patients were included in each type of analysis (NGS, mtDNA sequencing, microarray, etc.). Why do authors study patients only without a family history or syndromic disease?

Why GUSB gene was used as a reference for RT-qPCR?

I think that sections within the “Results” should be titled based on the result obtained instead of the type of analysis.

Authors performed unsupervised clusterization based on Pheo-Type gene expression profiling (Fig.1) but also presented genotypic subtypes. How these subtypes were determined? I recommend restructuring the “Results” and first describing gene mutation analysis.

How many tumors were included in the unsupervised clusterization?

The goal of the Swedish and French cohort comparison using only transcriptome data is not obvious. Mutations in susceptibly genes, which are driver events in paraganglioma development, should be taken into account.

Figure S1 presents a heatmap of top50 differentially expressed genes and unsurprisingly that these gene set divide two cohorts into two clusters. Moreover, genes were not listed on the heatmap.

“A next-generation sequencing panel for PCCs/PGLs susceptibility genes”

How many patients were included in the mutation analysis?

One patient was found to have a mutation in only one gene studied?

Table 1:

- what are bold letters indicate?

- add the link where the variant marked as “yes (?)” was reported.

- what does sporadic paraganglioma (†) mean in the table? As I understand all tumors studied were sporadic (without a family history).

- change column's name “Presence in normal tissue” to, for example, “Mutational status” (germline/somatic).

- add info on the variant pathogenicity, some of them are US (e.g. Ile412Asn).

What mean the titles of identified mtDNA haplogroups (HV/H/V, J/T, U/K/, and others) (Table S3)? Where are CI and p-value for HV/H/V haplogroup? P-values for other haplogroups do not pass the threshold p less than 0.05.

Two phrases, “…21 were novel synonymous (1), missense (13), nonsense (2), frameshift (1), promotor (1), tRNA (2) or rRNA (1) mutations …” and “The screening of genetic variants in mitochondrial genes revealed eight novel missense mutations including three germline mutations”, tell about the different number of identified novel missense mutations (13 and 8).

“The mtDNA contents of the 77 pheochromocytomas and normal adrenal medulla..” – according to materials and methods 74 PCCs and 3 PGLs were used.

“Gene expression analysis of the mitochondrial OXPHOS genes showed that there were no differences in mtRNA levels in the tumors presenting novel mitochondrial mutations compared to the rest of PCCs/PGLs, in the corresponding mutated genes” – Why was expression compared for genes with new and known mutations?

“….mitochondrial OXPHOS proteins compared to the corresponding normal tissue and other tumor tissues, although the observed differences were not statistically significant” – this is redundant information if the correlation is not statistically significant.

“We observed that the mtOXPHOS mRNA levels, generally are upregulated across the Pheo-Type based gene expression signatures” – According to Figure 1, almost all mtOXPHOS-related genes were upregulated in all samples and this has no association with Pheo-Type based gene expression signatures.
